# Sesquiterpene Lactones with Dual Inhibitory Activity against the *Trypanosoma brucei* Pteridine Reductase 1 and Dihydrofolate Reductase

**DOI:** 10.3390/molecules27010149

**Published:** 2021-12-27

**Authors:** Katharina Possart, Fabian C. Herrmann, Joachim Jose, Maria P. Costi, Thomas J. Schmidt

**Affiliations:** 1Institute of Pharmaceutical Biology and Phytochemistry (IPBP), University of Muenster, PharmaCampus, Corrensstrasse 48, D-48149 Muenster, Germany; k_poss01@uni-muenster.de (K.P.); f_herr01@uni-muenster.de (F.C.H.); 2Institute of Pharmaceutical and Medicinal Chemistry, University of Muenster, PharmaCampus, Corrensstrasse 48, D-48149 Muenster, Germany; joachim.jose@uni-muenster.de; 3Department of Life Sciences, University of Modena and Reggio Emilia, Via G. Campi 103, 41125 Modena, Italy; mariapaola.costi@unimore.it

**Keywords:** *Trypanosoma brucei*, human African trypanosomiasis, pteridine reductase 1 inhibitor, dihydrofolate reductase inhibitor, sesquiterpene lactones, natural products, in silico screening

## Abstract

The parasite *Trypanosoma brucei* (*T*. *brucei*) is responsible for human African trypanosomiasis (HAT) and the cattle disease “Nagana” which to this day cause severe medical and socio-economic issues for the affected areas in Africa. So far, most of the available treatment options are accompanied by harmful side effects and are constantly challenged by newly emerging drug resistances. Since trypanosomatids are auxotrophic for folate, their pteridine metabolism provides a promising target for an innovative chemotherapeutic treatment. They are equipped with a unique corresponding enzyme system consisting of the bifunctional dihydrofolate reductase-thymidylate synthase (*Tb*DHFR-TS) and the pteridine reductase 1 (*Tb*PTR1). Previously, gene knockout experiments with PTR1 null mutants have underlined the importance of these enzymes for parasite survival. In a search for new chemical entities with a dual inhibitory activity against the *Tb*PTR1 and *Tb*DHFR, a multi-step in silico procedure was employed to pre-select promising candidates against the targeted enzymes from a natural product database. Among others, the sesquiterpene lactones (STLs) cynaropicrin and cnicin were identified as in silico hits. Consequently, an in-house database of 118 STLs was submitted to an in silico screening yielding 29 further virtual hits. Ten STLs were subsequently tested against the target enzymes in vitro in a spectrophotometric inhibition assay. Five compounds displayed an inhibition over 50% against *Tb*PTR1 as well as three compounds against *Tb*DHFR. Cynaropicrin turned out to be the most interesting hit since it inhibited both *Tb*PTR1 and *Tb*DHFR, reaching IC_50_ values of 12.4 µM and 7.1 µM, respectively.

## 1. Introduction

The genus *Trypanosoma* contributes to the wide range of human-pathogenic parasites that are responsible for tropical diseases, of which 20 are still classified as neglected tropical diseases (NTDs) by the World Health Organization [1]. Although NTDs occur in 149 countries and put more than one billion people at risk, the available treatment options are still widely insufficient. In addition to the high mortality rates, the consequences of the diseases result in social stigmatisation, severe physical impairment of those affected and promote poverty in the regions involved [2,3].

The unicellular eukaryotic blood parasite *Trypanosoma brucei* (*T*. *brucei*) is the cause of human African trypanosomiasis (HAT, sleeping sickness) as well as the cattle disease “Nagana”, which are transmitted by the tsetse fly (*Glossina* spp.). Both illnesses, caused by different subspecies of *T. brucei*, still represent a massive health burden for many countries in Africa. HAT caused a variety of epidemiological outbreaks over the past century, which have only started to decline in the past 20 years. Since the documentation of 33,000 cases in the year 2000, the numbers have dropped below 1000 in 2019, which gives hope for the elimination of HAT as a public health concern. This progress has been accelerated by the recent introduction of the oral drug fexinidazole through the efforts of the Drugs for Neglected Diseases Initiative (DNDi) and Sanofi. Its easy application and manageable side effects differ significantly from the other medication options available, which are accompanied by severe adverse reactions and require hospitalization as well as constant medical supervision [4,5,6].

To prevent another relapse in the eradication of HAT, multiple factors need to be considered: possible political or social instabilities of the affected areas and environmental shifts as well as occurring epidemics of other illnesses could lead to increased infection rates [7,8]. One of the biggest challenges is the highly adaptive nature of trypanosomes, associated with their ability for immune evasion, that has so far prevented the successful development of a vaccine [9]. On top of that, the parasites occasionally exchange genetic information via sexual intercourse in the salivary glands of their vector, which increases virulence and drug resistance [10]. The existing and foreseeable resistance developments call for a constant addition of new drugs to the available treatment options, so that the search for new chemical entities with antitrypanosomal activity remains an important goal.

In the past years, the trypanosomal pteridine metabolism has been explored as a new drug target in the development of new chemotherapeutics against HAT. The family Trypanosomatidae developed a pteridine auxotrophy due to reductive evolution and became dependent on extracellular uptake of pteridines/folates. The corresponding enzymes of the bifunctional dihydrofolate reductase-thymidylate synthase (DHFR-TS) and the pteridine reductase 1 (PTR1) play a key role in the metabolization of pteridines and consequently in cell survival [11]. Therefore, the dual inhibition of the mentioned enzymes would be highly desirable for an anti-folate drug acting against Trypanosomatids [12].

The DHFR-TS enzyme consists of an N-terminal DHFR domain that connects to the C-terminal TS domain via a linker peptide. Past studies of protozoan DHFR-TS indicated that the coupling of both enzymes enables substrate channelling between the domains, creating a tighter control of metabolic flux and a survival advantage for the parasites [13]. The DHFR is an oxidoreductase that selectively reduces 7,8-dihydrofolate (DHF) to 5,6,7,8-tetrahydrofolate (THF) using NADPH as co-substrate. Catalysed by the glycine cleavage system (GCS), THF can be transformed into 5,10-methylenetetrahydrofolate (CH_2_THF), which serves as a C1-donor during the reductive methylation of deoxyuridine monophosphate (dUMP) to deoxythymidine monophosphate (dTMP) by the TS. This process secures the sufficient supply of deoxythymidine for DNA biosynthesis [14].

The NADPH-dependent short-chain dehydrogenase-reductase PTR1 is an enzyme unique to trypanosomatidae that ensures the metabolization of folates and other pteridines in conjunction with DHFR. Under normal physiological conditions, it mainly catalyses the reduction of its primary substrate pterin to tetrahydrobiopterin and only transforms 10% of folate to THF. In the case of DHFR inhibition, the expression of PTR1 is upregulated to provide THF for the upkeep of the folate metabolism. Furthermore, gene knockout experiments with *Tb*PTR1 null mutants resulted in a loss of virulence and viability in cell cultures as well as animal models, highlighting the major significance of this enzyme for parasite survival [15].

Due to the described metabolic bypass for THF synthesis, classic antifolates used in cancer therapy against bacterial infections and malaria have no sufficient antitrypanosomal or antileishmanial effect. Common drugs such as methotrexate, pyrimethamine and cycloguanil are not selective towards the trypanosomal DHFR and demonstrate insufficient inhibition of the PTR1 [16,17,18]. Hence, no antifolates have been used against HAT or other trypanosomatid diseases up to now [18]. The search for potent lead structures is of great importance to enable a dual inhibition of both enzymes by one or more agents and therefore enable the exploitation of the trypanosomal folate metabolism as a therapeutic target.

Past discoveries from natural product platforms have been an essential driving force in the treatment and management of tropical diseases. The vast diversity of compounds derived from plants, bacteria and marine organisms as well as fungi has been the fundamental pillar of traditional medicine and continues to provide inspiration for the design of modern drugs [19]. In this context, sesquiterpene lactones (STLs), a particular class of plant terpenoids with an impressive range of biological activities [20], have displayed very promising antitrypanosomal activity which has made some of them highly interesting lead scaffolds for the development of new drugs against HAT [21,22].

In this work, we used a pharmacophore-based virtual screening of natural product databases followed by an in vitro evaluation through spectrophotometric inhibition assays to identify compounds with inhibitory activity against recombinant *Trypanosoma brucei* dihydrofolate reductase (*Tb*DHFR) and pteridine reductase 1 (*Tb*PTR1). Various STLs were among the in silico hits and were consequently tested in vitro for inhibition of both enzymes.

## 2. Results

### 2.1. In Silico Investigation of Natural Products as Potential Inhibitors of TbPTR1 and TbDHFR

For the in silico simulation of protein–ligand interactions, five 3D protein structures of the *Trypanosoma brucei* pteridine reductase 1 (*Tb*PTR1) and two structures of the dihydrofolate reductase (*Tb*DHFR) were chosen from the Protein Data Bank (PDB), each containing a co-crystallized compound with an experimentally verified inhibitory activity as well as the co-substrate NADP. Using the software Molecular Operating Environment (MOE), a complex- as well as a target-based pharmacophore model was created for each structure, based on the particular features of the binding pocket and co-crystallized inhibitor of each PDB entry, resulting in 14 pharmacophore models (for details see Section 4.1.3.).

A natural product database containing 1100 commercially available natural compounds was filtered for drug-like substances applying Lipinski’s rule of five, resulting in 737 natural products. The ten most favourable low-energy conformers of each compound were subjected to a pharmacophore-based virtual screening (for details see Section 4.1.4.), followed by molecular docking simulations (for details see Section 4.1.5.). After applying an induced fit docking approach, the best ranking hits of the pharmacophore screening yielded two STLs as virtual hits among the wide range of diverse natural products: the germacranolide-type STL cnicin (**1**) (a constituent of *Centaurea benedicta* (L.) L. (synonym *Cnicus benedictus* L.), Asteraceae) appeared as a top hit for the *Tb*PTR1 protein model “4CMK” and the guaianolide-type STL cynaropicrin (**2**) from artichoke (*Cynara cardunculus* L., Asteraceae) for the *Tb*DHFR model “3RG9”. To illustrate the rather favourable predicted binding of STLs **1** and **2** to the enzymes under study, their highest scoring docking poses in the respective substrate binding pockets are shown in Figure 1 and Figure 2.

Figure 1A depicts the energetically most favourable docking conformation of cnicin in the *Tb*PTR1 binding pocket (model “4CMK”, S-score = −8.69 kcal/mol). The protein–ligand complex generated by MOE shows the voluminous germacrene scaffold fitting comfortably near the predominantly lipophilic rim of the catalytic cavity formed by Phe171 and further amino acids. The ester side chain attached to C-6 of **1** is oriented towards the centre of the binding pocket, its diol substructure forming hydrogen bonds with the oxygen atoms of the NADP phosphate linker. Furthermore, the carboxyl group of the lactone substructure acts as an H-bond acceptor for the thiol proton of Cys168.

Figure 1B portrays the top docking conformation of cynaropicrin in the *Tb*DHFR catalytic side (model “3RG9”, S-score = −7.79 kcal/mol). The guaianolide ring system is located in the lipophilic hollow near the border of the binding pocket. The software MOE postulates a carbon atom in position 6 as well as the hydroxyl group acting as H-bond donors for the sulfide of the closely located Met55.

Following the in silico prediction that cnicin and cynaropicrin are inhibitors of the enzymes under study, an in house collection of 118 STLs of various structural subclasses from our previous work [22] was also submitted to the pharmacophore-based virtual screening and docking procedure as outlined above. The top hits from the docking simulations are depicted in Appendix A for *Tb*PTR1 and Appendix A for *Tb*DHFR (Appendix A), respectively. Overall, 29 further compounds displayed favourable docking scores against one or both enzymes. Based on their availability, eight of these compounds were hence chosen, along with **1** and **2**, to be tested for their potential inhibitory activity against the two enzymes in vitro.

### 2.2. In Vitro Evaluation of the In Silico Hits against TbDHFR and TbPTR1

Both *Tb*DHFR and *Tb*PTR1 were obtained by recombinant expression in *E. coli* and enzyme inhibition assays established based on previous reports [23,24,25].

The ten selected STLs identified as hits in silico (structures see Appendix A; the structures of the remaining 21 in silico hits are shown in Appendix A) were initially tested in vitro by determining their relative inhibitory potency at a set concentration (% inhibition of activity at 100 µM for *Tb*PTR1; 50 µM for *Tb*DHFR) using recombinant *Tb*PTR1 and *Tb*DHFR. In cases where the relative inhibition thus determined was >50%, the half maximal inhibitory concentration (IC_50_) or half maximal effective concentration (EC_50_) values of the compounds were determined by recording concentration–effect curves (for details see Section 4.9 and Appendix A). The determined relative inhibition as well as the respective IC_50_ or EC_50_ values are listed in Table 1 for both investigated enzymes.

All ten STLs tested in vitro (Figure 2) showed more than 10% relative inhibition of *Tb*PTR1 at a concentration of 100 µM (hit rate = 100%), five of them inhibiting over 50% of the enzyme activity and thus qualifying for determination of IC_50_ values. Due to solubility issues, IC_50_ values could not be determined in the case of the germacranolide-type STLs cnicin (**1**) as well as (*Z*)-eucannabinolide (**5**), so EC_50_ values are reported for these compounds.

Eight of the ten compounds that were tested against *Tb*DHFR in vitro were active against the enzyme at 50 µM (hit rate = 80%). In this case, three STLs were able to inhibit *Tb*DHFR over 50%.

The STLs **1**–**5** displayed more than 50% inhibition of *Tb*PTR1, with compound **2** achieving the lowest IC_50_ value at 12.4 µM. Of the five more potent *Tb*PTR1 inhibitors, compounds **2** (IC_50_ = 7.1 µM) and **3** (IC_50_ = 13.3 µM) also inhibited the *Tb*DHFR over 50%, while compounds **1** and **4** showed moderate to low activity against this enzyme. No *Tb*DHFR inhibition could be detected for compound **5**. Cynaropicrin (**2**) hence turned out as the most potent dual inhibitor of the compounds under study.

## 3. Discussion

In the present work, we aimed for the identification of inhibitors of the pteridine-metabolizing enzymes *Tb*PTR1 and *Tb*DHFR to interrupt the folate and pteridine metabolism of *T*. *brucei*. In *T*. *brucei*, the correct function of the *Tb*PTR1 alone is indispensable for the upkeep of pteridine metabolism so that *Tb*PTR1 inhibitors can be useful as new drug leads, whereas the inhibition of *Tb*DHFR alone would not be sufficient to fight the parasite in an efficient manner [15]. However, dual inhibitors affecting both enzymes simultaneously can be conceived to be more efficient trypanocides than compounds inhibiting either one of these enzymes, so that we included both enzymes in the present in silico and in vitro study.

Natural products have in many cases shown very promising activity against protozoan parasites, including *T. brucei* [30,31]. Our group has therefore continuously searched for new antiprotozoal compounds of natural origin. Thus, the present study was initially based on a chemically diverse library of commercially available natural products. As part of the results emerging from our group’s work in the past, various natural products from the class of sesquiterpene lactones (STLs) were found to represent very interesting hits against *T. brucei* [32]. Extensive structure–activity relationship studies were carried out over the years to explain and distinguish the antiprotozoal effects of STLs [21,22,33,34]. Due to their very prominent activity against *T. brucei*, some STLs represent an interesting substance class for the lead discovery against HAT. Notably, two STLs (**1** and **2**) were found among the in silico hits of the initial virtual screening (VS) of the present study. Therefore, the STLs of our in-house library were also subjected to the VS protocol against the enzymes under study and a variety of 29 further in silico hits were thus found.

The α-methylene-γ-lactone group characteristic to most sesquiterpene lactones contains a reactive α, β-unsaturated carbonyl group. This Michael acceptor is a strong alkylating agent of bionucleophiles (especially SH groups) and has been proven essential for many biological activities of STLs, including their antitrypanosomal potential [21,22,30,33,34,35]. Even though it cannot be proven experimentally at present, it appears conceivable that formation of a covalent bond with Cys168 in the catalytic centre of *Tb*PTR1 could be involved in the inhibitory activity of some STLs on this enzyme. Even though the force field-based docking protocol used in this study does not simulate formation of covalent bonds, the vicinity of the thiol group of Cys168 and the reactive exomethylene group in the best docking poses of cnicin (**1**; compare Figure 1) and some other STLs would indicate that such an alkylation may be possible. Experimental investigations on the inhibition kinetics of **1** and **2** will have to be conducted to corroborate this hypothesis. However, the noticeable difference in the activities of the various tested STLs—all equipped with a reactive structure element of this type—suggest that their specific affinity toward the target enzymes is not based on the putative covalent interaction of the reactive α-methylene-γ-lactone groups alone. The relatively large lipophilic parts of the sesquiterpene lactones’ core scaffolds would be favourable for strong hydrophobic interactions with the predominantly nonpolar areas of the binding pockets of both *Tb*PTR1 as well as *Tb*DHFR [20]. However, the intensity of possible covalent interactions of α, β-unsaturated substructures with amino acids of the binding pocket will also depend strongly on steric feasibility, i.e., the overall geometry that the STL can adopt within the restrictions of the binding site on the protein. Additionally, the propensity for further stabilizing interactions such as hydrophobic, van der Waals and H-bonding interactions are important for deciding whether an STL efficiently inhibits the enzymes under study. Thus, only STLs **1**–**5** showed strong inhibitory activity against *Tb*PTR1 (>50% at 100 µM), and only compounds **2**, **3** and **9** showed potent activity against *Tb*DHFR (Table 1).

We demonstrated that out of the investigated sesquiterpene lactones, compounds **2** and **3** are dual inhibitors, affecting both enzymes with similar potency. Furthermore, **1** and **4** also inhibit both enzymes, but their effect against *Tb*DHFR is much weaker than that against *Tb*PTR1. It is noteworthy that all the identified compounds had previously been shown to inhibit the growth of *T. brucei* rather potently in phenotypic cellular assays, with IC_50_ values between 0.1 and 4 µM. These IC_50_ values of antitrypanosomal activity are well below those found with the isolated enzymes in the course of this study. Furthermore, it becomes clear that STLs such as, e.g., **10**, which do not show particularly strong activity against either enzyme, may still possess considerable antitrypanosomal activity. These findings clearly indicate that inhibition of these enzymes is not likely to represent the sole mechanism of action of the identified hits.

Most notably, cynaropicrin **2** from Artichoke (*Cynara cardunculus*, Asteraceae) [26], representing the strongest dual inhibitor of the present study with IC_50_ of 12.4 µM against *Tb*PTR1 and 7.3 µM against *Tb*DHFR, is known to be a very potent inhibitor of *T. brucei* growth (IC_50_ = 0.3 µM) and has also been proven to possess in vivo activity against the parasite in an acute mouse model [36]. The antitrypanosomal mechanism of action of **2**—similar to that of other STLs—is not fully understood. It has been connected with the trypanothione redox system in *T*. *brucei* [37]. Our present study reveals the pteridine metabolism as a second target of this highly interesting natural compound in *T. brucei*, offering further explanation for its impressive antitrypanosomal effect.

The 9β-hydroxyparthenolide-ester **3** (*Inula montbretiana*, Asteraceae) had also demonstrated to have quite potent antitrypanosomal activity in *T*. *brucei* cell assays (IC_50_ = 1.3 µM) in the past [27] and, in the present study, exhibited potent inhibitory activity against both enzymes, although it had a stronger affinity towards *Tb*DHFR (IC_50_ = 13.3 µM) compared with *Tb*PTR1 (IC_50_ = 40.5 µM). In addition to **2**, it may hence be another interesting candidate for further studies.

In conclusion, the STLs identified in this study, especially the dual inhibitors **2** and **3**, as disruptors of a parasite-specific vital metabolic process in *T*. *brucei*, represent a promising starting point for structure optimization aiming at even more potent inhibitors of *T. brucei* pteridine metabolism. The newly gained knowledge about compounds **2** and **3** belonging to the few known compounds with dual inhibitory activity against *Tb*PTR1 and *Tb*DHFR can be used in the search for inhibitors of related enzyme systems in other human pathogenic trypanosomatids in the future. The present results, furthermore, add an interesting aspect to the knowledge on the antitrypanosomal mechanism(s) of action of STLs.

Finally, it must be mentioned that the initial virtual screening of the natural product library yielded a variety of further non-STL hits, which are currently under study and will be the subject of a subsequent communication.

## 4. Materials and Methods

### 4.1. In Silico Procedure

All in silico experiments were carried out using the software Molecular Operating Environment (MOE, Chemical Computing Group, Montreal, QC, Canada) v. 2018.0101 applying the integrated force field MMFF94x (Merck Molecular Force Field) [38].

#### 4.1.1. Preparation of the Respective 3D Protein Structures

To allow in silico predictions about possible interactions of natural products with the enzymes under study, *Tb*PTR1 and *Tb*DHFR, 3D protein structures of the enzymes were retrieved from the Protein Data Bank (PDB) of the Research Collaboratory for Structural Bioinformatics (RCSB) [39]. Five 3D models were chosen in case of *Tb*PTR1 (PDB-IDs “2 × 9G”, “3MCV”, “4CMJ”, “4CMK” and “5JDI” [40,41,42,43]) as well as two models of *Tb*DHFR (PDB-IDs “3QFX” and “3RG9” [44,45]). Initially, incomplete or missing terminal amino acids of the protein models that may arise from the X-ray crystallographic data were detected and corrected by the software (MOE: *Compute* → *Prepare* → *Structure Preparation* → *Correct*). Next, the enzyme model was virtually titrated at a set temperature and pH (27 °C, pH 7.0) to adjust the ionisation stage of basic and acidic amino acid side chains and assign the hydrogen atoms accordingly (MOE: *Compute* → *Prepare* → *Protonate 3D*). Subsequently, an energy minimization was carried out to relax the protein structure within the conditions of the assigned force field. To prevent a significant change in the experimentally determined protein structure, this process was performed incrementally by fixing the positions of all heavy atoms, only allowing gradual movement within a radius of 0.5 to 1.5 Å (MOE: *Compute* → *Energy Minimize; Atoms: Tether*). Finally, an energy minimization without constraints was performed, leading to fully relaxed protein structures for further study.

All following experiments were carried out with the 3D models thus optimized, including their respective co-crystallized inhibitors and co-substrate NADP.

#### 4.1.2. Natural Product Databases

In this work, a virtual library containing 3D structures of diverse natural products, commercially available from Phytolab GmbH (Vestenbergsgreuth, Germany), was considered for virtual screening.

The “Rule of Five” by Lipinski et al. was applied as a filter to narrow down the databases to drug-like compounds. The descriptor “*lip-druglike*” returns a value of 1 if more than one violation with Lipinski’s rule is noted; compounds with a descriptor value > 0 were hence omitted [46]. After generating their 3D structures with MOE, all remaining 737 drug-like compounds in the database compounds underwent a low mode molecular dynamics conformational search (*LowModeMD*) to determine the most favourable conformations for each natural product. An energy threshold of 3 kcal/mol above the lowest energy conformer was applied, and the conformation limit was set to ten for each compound. The resulting conformers were collected in separate databases and used for the following virtual screening process.

Following the observation that two sesquiterpene lactones from this database (cnicin (**1**) and cynaropicrin (**2**)) were predicted in silico to be inhibitors (hits) and proven active against the enzymes in vitro, an in-house virtual library from our previous studies [34] consisting of 120 sesquiterpene lactone structures was filtered according to Lipinski’s rule, with 118 STLs remaining for further investigation.

#### 4.1.3. Pharmacophore Design

Aiming to limit and refine the selection of test compounds which would undergo further in silico studies (docking simulations), two different pharmacophore hypotheses were created for every protein structure: (a) a complex-based pharmacophore to define the crucial interactions of the co-crystallized inhibitor with the nearby amino acids of the catalytic cavity and (b) a mere target-based pharmacophore wherein only the structure and properties of the binding pocket were taken into consideration. Possible interactions with the co-crystallized co-substrate NADP were taken into account during both strategies.

All interactions detected by the software were computed and visualized (MOE: *Compute* → *Ligand Interactions*). Each supramolecular interaction with a calculated energy of ≤−1 kcal/mol was ranked accordingly and represented as a feature sphere in the 3D model (MOE: *New* → *Pharmacophore Query* → *Feature*). Each sphere contained information about the type of interaction and its estimated radius (radius between 1.0 and 1.4 Å, as suggested by MOE). All postulated interactions were reviewed, and missing interactions were integrated by hand if necessary. Depending on the model, between 5 and 20 feature spheres were created. To eliminate compounds that would collide with the protein binding pocket due to their geometry or size, so-called exclusion spheres were drawn around each atom of the protein and co-substrate, excluding the solvent and ligand molecules. The volume of the spheres was adjusted individually to avoid collisions with the co-crystallized inhibitor. The pharmacophores were considered validated if the co-crystallized inhibitor could be identified using the complete set of feature spheres during a screening. All complex- and target-based pharmacophore models are presented in the Appendix A.

#### 4.1.4. Virtual Screening

The obtained pharmacophores were used as a 3D query to filter the natural product libraries during a virtual screening (MOE: *Pharmacophore Editor* → *Search; Results: Conformations; Hits: Best Per Molecule*). Initially, all feature spheres were included. If this resulted in the identification of fewer than ten matching compounds from the screened database, the structural requirements for the substances were gradually reduced by decreasing the number of spheres according to their ranking until 10 to 120 hits were obtained (MOE: *Pharmacophore Editor, Partial Match: All but*).

#### 4.1.5. Docking Simulations

The hits obtained from the virtual screening were subjected to a two-step docking simulation to predict their preferred orientation within the binding pocket of each enzyme model (MOE: *Compute* → *Dock*).

The calculated Gibbs energy (∆G) arising from the generated protein–ligand complexes (docking poses) was used as a scoring parameter (MOE: *London dG*). The docking conformations were transferred to a separate database along with their S-scores. The S-score corresponds to the simulated free energy in kcal/mol of each complex. Thus, a more negative value implies a spontaneous reaction; therefore, all docking poses were ranked by ascending S-score.

To validate each docking simulation and to obtain an estimate for the binding affinity (S-scores) in comparison with a known inhibitor, a so-called self-dock of the co-crystallized inhibitors with their corresponding protein model was performed in each case using the induced fit mode. The poses generated from the self-dock were compared with the experimentally obtained conformation and generally yielded binding modes and orientations very similar to the experimental ones. The lowest S-score from each self-dock was selected for each 3D protein structure and used as reference for all following docking simulations with that particular protein model.

The docking-based screening was performed in two stages: During the first stage of the docking process, only the low molecular natural products were placed inside the catalytic cavity in different conformations or tautomers, but the protein structure was not treated as a flexible entity (MOE: *Rigid Receptor*). The ten top docking poses obtained by this procedure were subjected to a second round of docking simulation during which the possible conformations of the compound as well as the amino acids of the binding pocket were simulated flexibly (MOE: *Induced fit*). The five top hits from the induced fit docking with each pharmacophore model qualified to be tested experimentally in spectrophotometric inhibition assays. Based on their availability, ten of these STLs were chosen for experimental evaluation.

### 4.2. Recombinant Expression and Purification of TbPTR1

For the heterologous expression of the *Trypanosoma brucei* PTR1, an already transformed *E. coli* BL21 (DE3) strain was used, kindly provided by the working group of Prof. Dr. M. Paola Costi (Modena, Italy). The strain contained the plasmid pET-15b::*Tb*PTR1His as vector, encoding for *Tb*PTR1 controlled by the T7/lac promotor as well as an N-terminal His6-Tag and a carbenicillin resistance gene [23].

Recombinant *Tb*PTR1 was purified following a modified procedure by Sambrook and Russell [25,45]. An overnight culture of the transformed *E*. *coli* BL21(DE3) cells was used to inoculate 2 L Erlenmeyer flasks filled with 400 mL LB medium (1:1000) and incubated until an optical density (OD_578nm_) of 0.6 to 0.9 was reached (4–6 h, 37 °C, 200 rpm). All cultures were supplemented with 50 µg/mL carbenicillin. The expression was induced by adding 0.4 mM isopropyl-*β*-D-thiogalactopyranoside (IPTG) to the culture, followed by incubation overnight (20 h, 20 °C, 200 rpm). The cells were harvested by centrifugation (4 °C, 10,000 rpm, 10 min) and resuspended in lysis buffer (50 mM Tris/HCl (pH 7.6), 250 mM NaCl), containing lysozyme (1 mg/mL), deoxyribonuclease I (DNase I, 1 mg/mL), phenylmethanesulfonyl fluoride (PMSF, 1 mM) and benzamidine (1 mM). Second, mechanical lysis was performed by sonication on ice. After recovering the soluble fraction of the lysate through two additional centrifugation steps (4 °C, 10,000 rpm, 10 min), recombinant *Tb*PTR1 was purified by immobilized metal ion affinity chromatography (IMAC) using a nitrilotriacetate (NTA-Ni^2+^) loaded column. Initially, the column was rinsed with washing buffer (50 mM Tris/HCl (pH 7.6) and 250 mM NaCl, 20 mM imidazole), followed by elution of the fusion protein in multiple fractions by adding buffers with increasing imidazole concentrations (100–500 mM). A 12.5% polyacrylamide gel electrophoresis (SDS-PAGE) was used to identify fractions containing the target protein, which were subsequently pooled and further purified through dialysis (50 mM Tris/HCl (pH 7.6), 100 mM NaCl) for 4 h at 4 °C. Lastly, the lysate was supplemented with 20% glycerol for cryo protection and stored in aliquots at −80 °C.

### 4.3. Cloning of TbDHFR into E. coli BL21(DE3) Host Strain

The genomic sequence coding for *Trypanosoma brucei brucei* DHFR-TS available in Genbank server (Gene ID: 3658761) was utilized to design a suitable vector for the overexpression of *Tb*DHFR in *E*. *coli*. The DNA fragment containing the *Tb*DHFR sequence was amplified with Phusion DNA polymerase (Thermo Fisher Scientific, Bonn, Germany) using the forward and reverse primers KP01 (5′-CACCATCACCATCATATGCTGAGTCTGACCCGTATTC-3′) and KP07 (5′-CAGCCGGATCCGTTATTCTCGCTGTTACGCGGAAC-3′) (Eurofins MGW Operon, Ebersberg, Germany). The backbone from pET-11D-kduD (Merck KGaA, Darmstadt, Germany) [47] was amplified with primers SB001 (5′-TAACGGATCCGGCTGCTAAC-3′) and MS41 (5′-ATGATGGTGATGGTGGTGCATG-3′) and the original template DNA removed by *Dpn*I digestion. The PCR products were loaded on a 1% agarose gel (110 V for 50 min) for electrophoretic separation, purified using the QIAquick Gel Extraction Kit (Qiagen, Hilden, Germany) and combined to a plasmid by In-Fusion HD EcoDry cloning kit (Clontech, Saint-Germain-en-Laye, France). The obtained plasmid was transformed into *E. coli* Stellar competent cells (Invitrogen) and grown on LB agar containing 100 µg/mL carbenicillin (6 h, 37 °C). Positive clones containing the insert genes were screened by PCR amplification and cultured in 10 mL LB medium overnight (37 °C, 200 rpm). The plasmid was purified using the innuPREP Plasmid Mini Kit (Analytik Jena, Jena, Germany) and analysed by Seqlab (Goettingen, Germany). The correct construct encoding *Tb*DHFR with an N-terminal His6-Tag and a carbenicillin resistance gene under the control of the T7/lac promotor was labelled pKP03 and transformed into *E. coli* strain BL21(DE3) for heterologous expression.

### 4.4. Recombinant Expression and Purification of TbDHFR

The expression and purification procedure for *Tb*DHFR was performed in a manner analogous to Section 4.2. During the resuspension process of the cell pellet, 2-mercaptoethanol (BME, 7 mM) was added to the lysis buffer to ensure a reducing environment. Again, mechanical lysis was performed by sonication on ice. The soluble fraction of the lysate was collected after centrifugation (Section 4.2.) and *Tb*DHFR was purified by IMAC. After rinsing the column with washing buffer (50 mM Tris/HCl (pH 7.6), 250 mM NaCl, 20 mM imidazole), the fusion protein was eluted by adding buffers with increasing imidazole concentrations (50 and 500 mM). A 12.5% SDS-PAGE was employed to monitor fractions containing the target protein, which were pooled and dialysed in reducing conditions (50 mM Tris/HCl (pH 7.6), 100 mM NaCl, 10 mM dithiothreitol (DTT)) for 4 h at 4 °C. The lysate was supplemented with 20% glycerol and stored in aliquots at −80 °C.

### 4.5. Kinetic Characterization of TbPTR1

The concentration and the activity of *Tb*PTR1, as well as the saturating conditions of its substrate and co-substrate were determined by monitoring the oxidation of the co-substrate NADPH to NADP^+^ via UV/Vis spectroscopy (Hitachi U-2900, Tokyo, Japan). All measurements were observed for 250 s at 340 nm and a constant temperature of 30 °C. The experiments were carried out as triplicates using buffer A (50 mM Tris/HCl (pH 7.6), 250 mM NaCl).

The concentration of *Tb*PTR1 was 3.23 mg/mL. Its specific activity was calculated to be 0.03 U/mg. All experiments were carried out using the saturating concentrations of folic acid (8 µM) and NADPH (150 µM). The diagrams for the determination of the saturating conditions are depicted in the Appendix A.

### 4.6. Kinetic Characterization of TbDHFR

Kinetic analysis of *Tb*DHFR was performed according to 4.5. All measurements were carried out as triplicates using buffer B (50 mM Tris/HCl (pH 7.6), 250 mM NaCl, 10 mM BME).

The concentration of *Tb*DHFR was 0.04 mg/mL with a specific activity of 38.1 U/mg. All experiments were carried out using the saturating concentrations of dihydrofolate (DHF, 50 µM) and NADPH (150 µM). The diagrams for the determination of the saturating conditions are depicted in the Appendix A.

### 4.7. Test Compounds

STLs **1** and **2** were generously donated by Phytolab GmbH (Vestenbergsgreuth, Germany). Compounds **3**–**10** were isolated, and their antitrypanosomal activity was tested in previous studies of our group [21,27,28,29].

### 4.8. Single Concentration Assays

To determine the relative inhibition of the in silico hits, an initial in vitro screening was carried out under saturating conditions: *Tb*PTR1 96.99 µg, NADPH 150 µM, folic acid 8 µM in buffer A; *Tb*DHFR 0.21 µg, NADPH 150 µM, DHF 50 µM in buffer B. The compounds were prepared as 10 mM stock solutions in DMSO and tested at a set concentration (100 µM against *Tb*PTR1; 50 µM against *Tb*DHFR; different concentrations had to be applied since *Tb*DHFR was found to be more sensitive to DMSO; while *Tb*PTR1 activity was not influenced by up to 1.5% DMSO, *Tb*DHFR did not tolerate more than 0.5% DMSO in the assay). All measurements were carried out as duplicates and compared with a reference containing no inhibitor. The reference measurements were used to determine the full catalytic activity of the respective enzyme (100%), and the relative catalytic activity of all following measurements including inhibitors was calculated accordingly.

### 4.9. Determination of the IC_50_ Values

The compounds that were able to inhibit the activity of *Tb*PTR1 or *Tb*DHFR ≥ 50% in the single concentration assays were further investigated by the determination of their half maximal inhibitory concentration (IC_50_), including at least five inhibitor concentrations. If a complete inhibition of the enzyme could not be achieved experimentally due to the given assay conditions, the half maximal effective concentration (EC_50_) was determined instead. All measurements were carried out as triplicates against a reference containing no inhibitor at saturating conditions (Section 4.8.). The documented enzyme activity was analysed by nonlinear regression using GraphPad Prism 8 (GraphPad Software Inc., La Jolla, CA, USA).

## Figures and Tables

**Figure 1 molecules-27-00149-f001:**
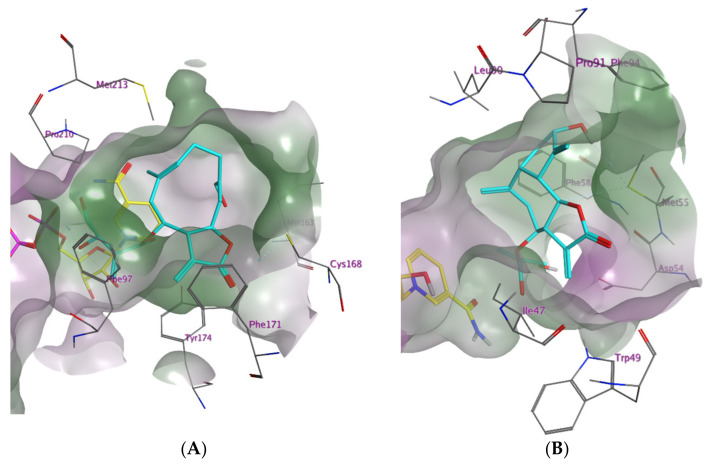
(**A**) Best scoring docking conformation for cnicin (**1**, carbon atoms coloured in cyan) in the binding pocket of *Tb*PTR1 (ID: „4CMK“) with co-crystallized NADP (carbon atoms coloured in yellow). The molecular surface is coloured according to lipophilicity with lipophilic areas in green and hydrophilic areas in purple. Co-crystallized solvent not shown; (**B**) Best scoring docking conformation for cynaropicrin (**2**, carbon atoms coloured in cyan) in the binding pocket of *Tb*DHFR (ID: “3RG9”) with co-crystallized NADPH (carbon atoms coloured in yellow). The molecular surface is coloured according to lipophilicity with lipophilic areas in green and hydrophilic areas in purple. Co-crystallized solvent not shown.

**Figure 2 molecules-27-00149-f002:**
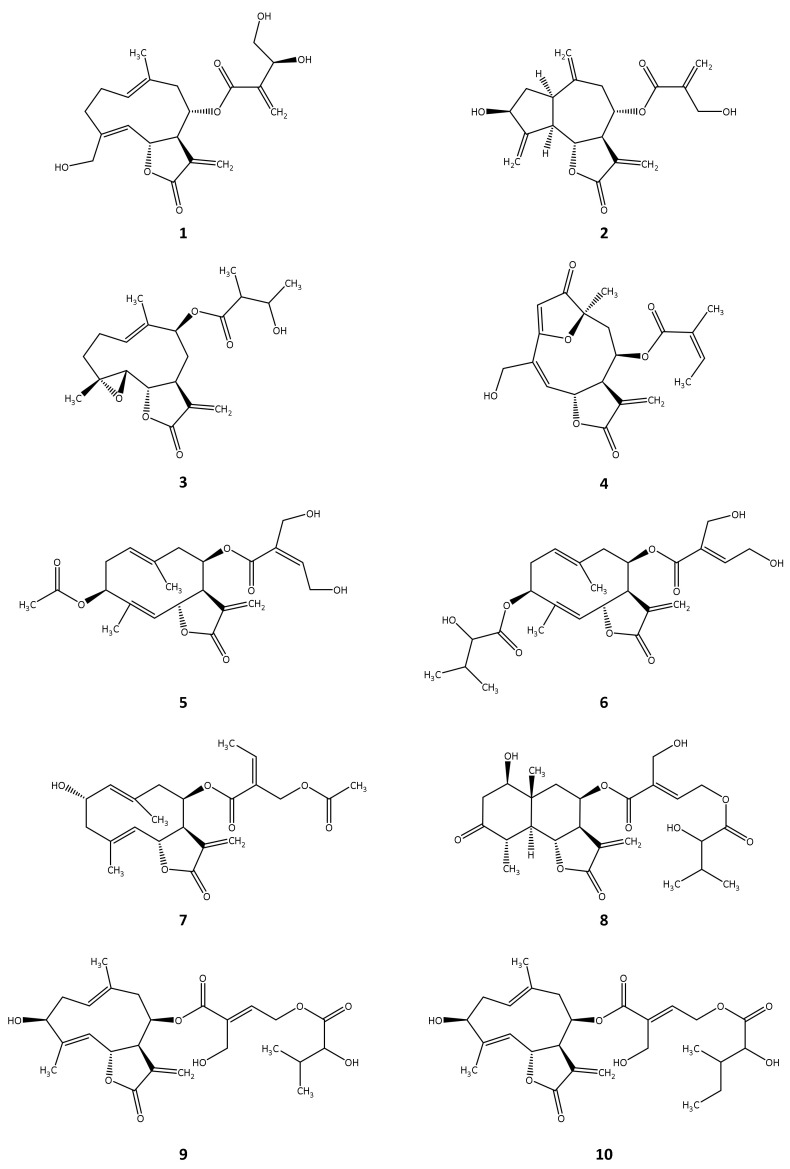
Chemical structures of the STLs tested in vitro.

**Table 1 molecules-27-00149-t001:** Inhibitory activity of the sesquiterpene lactones tested in vitro against *Tb*PTR1 and *Tb*DHFR. The IC_50_ values for in vitro growth inhibition of *T*. *brucei rhodesiense* as available in the literature are reported for comparison only.

Compound	*Tb*PTR1		*Tb*DHFR		*T*. *brucei* *rhodesiense*	
	Inhibition at 100 µM (%)	IC_50_ (µM)	Inhibition at 50 µM (%)	IC_50_ (µM)	IC_50_ (µM)	Reference
**1**	63.9	21.2 ^a^	15.2		0.4	[26]
**2**	92.1	12.4	95.8	7.1	0.3	[26]
**3**	73.1	40.5	63.8	13.3	1.3	[27]
**4**	79.9	30.5	34.9		0.1	[21]
**5**	58.1	31.5 ^a^	n.i.		1.7	[28]
**6**	33.1		n.i.		0.8	[28]
**7**	25.6		48.7		1.6	[29]
**8**	14.2		40.7		2.7	[28]
**9**	28.5		95.9	n.d.	2.6	[28]
**10**	26.7		20.1		4.0	[28]

^a^ EC_50_ values; n.i., no inhibition; n.d., not determined.

## Data Availability

All molecular modelling data as well as raw data of the enzyme inhibition study are available from the corresponding author on request.

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
