# Peer review of "Sesquiterpene Lactones with Dual Inhibitory Activity against the *Trypanosoma brucei* Pteridine Reductase 1 and Dihydrofolate Reductase"

_molecules, 2021, doi:10.3390/molecules27010149_

Round 1

Reviewer 1 Report

The study describes  a pharmacophore-based virtual screening of  sesquiterpene lactones followed by an in vitro evaluation of inhibitory activity against recombinant Trypanosoma brucei dihydrofolate reductase (TbDHFR) and pteridine reductase 1 (TbPTR1). The study has been well designed, executed and the results are very interesting and deserved to be published. These natural products were able to act as disruptors of a parasite-specific vital metabolic process in T. brucei and may lead to the design of more potent inhibitors of T. brucei pteridine metabolism. Although this study is very important for establishing a specific mode and mechanism of action, my only complaint about this study is the lack of data showing that these compounds do indeed display trpanocidal activity. There is a difference in inhibiting enzymes and actually being able to cross membranes, cell walls etc and arrive intact at the intended biological target. Inhibitory activity alone is not a guarantee that  useful antitrypanosomal activity will be realized.  It seems that the authors have already established the antitrypanosomal activity in previous studies   [references 21, 27–29]. I would suggest that these earlier results be included in discussion and correlated to the inhibitory activity disclosed in this report. Once this improvement has been made, I feel that the manuscript will be ready for publication.

Author Response

Reviewer 1

The study describes  a pharmacophore-based virtual screening of  sesquiterpene lactones followed by an in vitro evaluation of inhibitory activity against recombinant Trypanosoma brucei dihydrofolate reductase (TbDHFR) and pteridine reductase 1 (TbPTR1). The study has been well designed, executed and the results are very interesting and deserved to be published.

These natural products were able to act as disruptors of a parasite-specific vital metabolic process in T. brucei and may lead to the design of more potent inhibitors of T. brucei pteridine metabolism. Although this study is very important for establishing a specific mode and mechanism of action, my only complaint about this study is the lack of data showing that these compounds do indeed display trpanocidal activity.

The reviewer may have overlooked the data in the last column of Table 1 and the discussion of antitrypanosomal activity in the text.

There is a difference in inhibiting enzymes and actually being able to cross membranes, cell walls etc and arrive intact at the intended biological target. Inhibitory activity alone is not a guarantee that  useful antitrypanosomal activity will be realized.  It seems that the authors have already established the antitrypanosomal activity in previous studies   [references 21, 27–29]. I would suggest that these earlier results be included in discussion and correlated to the inhibitory activity disclosed in this report. Once this improvement has been made, I feel that the manuscript will be ready for publication.

In fact all these compounds are known to possess antitrypanosomal activity in in vitro, cynaropicrin even in vivo (mouse model), which is discussed in the manuscript (inhibition data reported in Table 1, last column, and thoroughly dicussed in section, lines 267 onward). The discussion was slightly extended as also suggested by reviewer 3.

We thank the reviewer for the very positive overall assessment and the time and effort spent to help us improve our work.

Reviewer 2 Report

This is well designed study with clear and solid data. However, please combine figure 1 and 2 as a single figure (e.g., Figure 1A and 1B). Please write the full name of all abbreviations where they appear first (both in abstract and main text). 

Author Response

Reviewer 2:

This is well designed study with clear and solid data. However, please combine figure 1 and 2 as a single figure (e.g., Figure 1A and 1B).

The figures were combined in one.

Please write the full name of all abbreviations where they appear first (both in abstract and main text). 

This was done.

We thank the reviewer for the very positive overall assessment and the time and effort spent to help us improve our work.

Reviewer 3 Report

In this manuscript, Possart et al performed in silico screening to identify inhibitors of pteridine metabolism followed by in vitro activity testing against the purified enzymes. Overall, this manuscript addresses an important issue in the neglected tropical disease field using appropriate methods. Nevertheless, I have the following major concerns:

  • IC50 values are very high for enzymatic assays, and much higher than the values required to kill brucei for these same compounds. This argues for a main mechanism of action on pathways other than pteridine metabolism. Text throughout the manuscript should be tempered accordingly, and this caveat mentioned in the abstract, results and discussion.
  • rule of 5 may not be appropriate for natural products (see Doak Bradley C, Over B, Giordanetto F, Kihlberg J. Oral Druggable Space beyond the Rule of 5: Insights from Drugs and Clinical Candidates. Chem Biol. 2014;21(9):1115-42). At minimum, the limitations of this filtering should be discussed.
  • please clarify why the initial activity assay was performed at different compound concentrations for the two enzymes

Minor issues:

  • line 265-266: it is inappropriate to describe compounds 1 and 4 as inhibiting both enzymes, given their very low percent inhibition at 50 µM on TbDHFR
  • line 281: it is inappropriate to describe any of the activities in this paper as potent, being in the µM range on purified enzymes.
  • many of the items marked as figures in the SI are actually tables (eg Fig S17-23) and should be re-labeled accordingly
  • Likewise, table S1 is actually a figure and should be re-labeled accordingly
  • structures displayed in Fig S17-23 are not explained. Please clarify.
  • line 178, remove the commas in “Both, TbDHFR and TbPTR1, were”.

Author Response

Reviewer 3

In this manuscript, Possart et al performed in silico screening to identify inhibitors of pteridine metabolism followed by in vitro activity testing against the purified enzymes. Overall, this manuscript addresses an important issue in the neglected tropical disease field using appropriate methods. Nevertheless, I have the following major concerns:

  • IC50 values are very high for enzymatic assays, and much higher than the values required to kill brucei for these same compounds. This argues for a main mechanism of action on pathways other than pteridine metabolism. Text throughout the manuscript should be tempered accordingly, and this caveat mentioned in the abstract, results and discussion.

The reviewer is certainly right. As another argument, various STLs (such as e.g. 10 in this study) that do not inhibit folate enzymes very significantly still possess considerable antitrypanosomal activity. This can be taken as an argument that inhibition of the enzymes under study is an additional mechanism of action shared by the compounds found here, but probably not the only mechanism of STLs; in case of cynaropicrin, we have already mentioned this, but maybe it was not clear enough. We have now tried to make it even clearer in the text by adding a more decisive explanation. (Lines 268 onwards in the new version).

  • rule of 5 may not be appropriate for natural products (see Doak Bradley C, Over B, Giordanetto F, Kihlberg J. Oral Druggable Space beyond the Rule of 5: Insights from Drugs and Clinical Candidates. Chem Biol. 2014;21(9):1115-42). At minimum, the limitations of this filtering should be discussed.

It is well established that natural products may be exceptions to the rule of 5 (Ro5). Nevertheless, we are not aware of any evidence that natural products do necessarily have to be such exceptions in all cases. The famous exception from the Ro5 has been established in order to account for the fact that very complex natural products with e.g. many H-bond donors or acceptors, high molecular mass etc. may still be orally available and feasible as drugs. Nevertheless, from a chemical standpoint, there is no general difference between natural and synthetic compounds. Thus, it appears an error of logic to expect that natural products should be exceptions per se (which would ultimately mean that no natural product should be expected to comply with the Ro5). In contrary, very many secondary metabolites from plants are rather druglike and do indeed comply very well with Lipinski’s rule (see e.g. Schmidt, T.J. Structure-Activity Relationships of Sesquiterpene Lactones. in: Atta-ur-Rahman (Ed.) Studies in Natural Products Chemistry, Vol. 33, pp 309-392 (2006) for Sesquiterpene Lactones).

On this background, we hope that the reviewer will agree that the application of the Lipinski filter can be useful in a context such as the present one and allow us to keep it without further comments on this issue which would, in our opinion, only make things more complicated.

  • please clarify why the initial activity assay was performed at different compound concentrations for the two enzymes

There is a simple technical explanation for this; DHFR does not tolerate DMSO conc. >0.5% (in contrast to PTR1 which tolerates concentrations up to 1.5%); therefore smaller volumes of our compound stock solutions had to be used in the initial measurements. This information has been added to the Material and Methods section (section 4.8, line 508 onward).

Minor issues:

  • line 265-266: it is inappropriate to describe compounds 1 and 4 as inhibiting both enzymes, given their very low percent inhibition at 50 µM on TbDHFR

We do mention that they are only weak inhibitors of DHFR so we do not see what we should change.

  • line 281: it is inappropriate to describe any of the activities in this paper as potent, being in the µM range on purified enzymes.

The term “potent” has been replaced by “significant”.

  • many of the items marked as figures in the SI are actually tables (eg Fig S17-23) and should be re-labeled accordingly

This has been corrected.

  • Likewise, table S1 is actually a figure and should be re-labeled accordingly

This has been corrected.

  • structures displayed in Fig S17-23 are not explained. Please clarify.

This has been clarified in the table captions.

  • line 178, remove the commas in “Both, TbDHFR and TbPTR1, were”.

This has been corrected.

We thank the reviewer for the positive overall assessment and the time and effort spent to help us improve our work.
